# A multi-scale study of thalamic state-dependent responsiveness

**Jorin Overwiening** [1,2]*, **Federico Tesler**[1], **Domenico Guarino**[1], **Alain Destexhe**[1]

**1** Department for Integrative and Computational Neuroscience, Paris-Saclay Institute of Neuroscience, Saclay, France, **2** Institute for Theoretical Physics, University of Muenster, Muenster, Germany

* alain.destexhe@cnrs.fr

**Data Availability Statement:** The numerical codes for simulating mean-field and spiking network and the code for fitting the transfer function of the mean-field on single cell data are openly available

## Abstract

The thalamus is the brain's central relay station, orchestrating sensory processing and cognitive functions. However, how thalamic function depends on internal and external states, is not well understood. A comprehensive understanding would necessitate the integration of single cell dynamics with their collective behavior at population level. For this we propose a biologically realistic mean-field model of the thalamus, describing thalamocortical relay neurons (TC) and thalamic reticular neurons (RE). We perform a multi-scale study of thalamic responsiveness and its dependence on cell and brain states. Building upon existing single-cell experiments we show that: (1) Awake and sleep-like states can be defined via the absence/presence of the neuromodulator acetylcholine (ACh), which indirectly controls bursting in TC and RE. (2) Thalamic response to sensory stimuli is linear in awake state and becomes nonlinear in sleep state, while cortical input generates nonlinear response in both awake and sleep state. (3) Stimulus response is controlled by cortical input, which suppresses responsiveness in awake state while it 'wakes-up' the thalamus in sleep state promoting a linear response. (4) Synaptic noise induces a global linear responsiveness, diminishing the difference in response between thalamic states. Finally, the model replicates spindle oscillations within a sleep-like state, exhibiting a qualitative change in activity and responsiveness. The development of this thalamic mean-field model provides a new tool for incorporating detailed thalamic dynamics in large scale brain simulations.

## Author summary

The thalamus is a fascinating brain region that acts as the gate for information flow between the brain and the external world. While its role and importance in sensory and motor functions is well-established, recent studies suggest it also plays a key role in higher-order functions such as attention, sleep, memory, and cognition. However, understanding how the thalamus acts on all these functions is challenging due to its complex interactions at both the neuron level and within larger brain networks. In this study, we used a mathematical model grounded in experimental data that realistically captures the behavior of the thalamus, connecting the scales of individual neurons with larger populations. We found that the thalamus functions differently depending on whether the brain is

at the corresponding GitHub page: https://github.com/joverwiening/thalamic-mean-field.

**Funding:** Research supported by the CNRS and the European Union (Human Brain Project H2020-945539 to AD, Virtual Brain Twin project Horizon Health 101137289 to AD) which paid the salary of JO, FT, DG, AD. The funders had no role in study design, data collection and analysis, decision to publish, or preparation of the manuscript.

**Competing interests:** The authors have declared that no competing interests exist.

in an awake or a sleep-like state: When awake, the thalamus processes sensory information in a straightforward way, resulting in a faithful information transmission to the cortex. But during sleep, only significant or important stimuli create a response. Importantly, this behavior can be controlled by cortical-like input and noise. With this study, we shed light on how the thalamus might modulate and interact with various brain functions across different scales and states. This research provides a deeper understanding of the thalamus's role and could inform future studies on sleep, attention, and related brain disorders.

## Introduction

The thalamus, a well preserved structure found in all mammals [1], serves as the core relay hub of the central nervous system. Diverse thalamic nuclei function as transmitters of sensory information from the periphery to the cortex and other central nervous system structures, while also facilitating the transfer of motor commands from the cortex to various regions of the body [2, p. 4–5]. Each of the relatively independent thalamic nuclei comprises at least two cell types: excitatory (glutamergic) principal *relay* cells, featuring extensive axonal projections to various nervous system structures, but rarely to other principal cells, and local inhibitory (GABAergic) interneurons [3, 4].

The primary source of activity in thalamic nuclei arises from direct pathways, operating in both peripheral-to-central and central-to-peripheral directions. Additionally, cortical feedback projections exert a strong influence on the thalamus. Notably, the number of thalamo-cortical outgoing axons is approximately one-tenth of the number of cortico-thalamic incoming axons [4, 5], and the cortex is the major source of synapses within the thalamus, for example accounting for 50% of synapses in the lateral geniculate nucleus (LGN) [4]. This extensive feedback loop between the thalamus and cortex indicates a substantial modulating role of the cortex in thalamic relay functions [6].

During attentive wakefulness, thalamic relay neurons display tonic firing. However, membrane hyperpolarization leads to bursting behavior via low-threshold $Ca^+$ channels [7]. Bursting occurs in deep sleep states (NREM) and general states of low attention [8], in which hyperpolarization is generated by a low level of the neuromodulator acetylcholine (ACh) [9].

Surrounding the thalamus, the thalamic reticular nucleus (TRN) contains GABAergic reticular cells (RE) that broadly inhibit thalamic nuclei through axonal, and themselves through dense axonal and dendritic connections [4, 7]. RE neurons can be activated through feedforward signals from thalamic nuclei or feedback from the cortex. These neurons consistently exhibit bursting behavior and can induce similar patterns in thalamic relay cells via hyperpolarization. This recurrent network allows the cortex and thalamus itself to actively modulate thalamic response and transfer of information, rendering the thalamus as a gate, with the TRN as the gatekeeper.

In addition to its gating function, there is also evidence of the thalamus playing a principal role in whole brain dynamics, such as spindle oscillations or slow waves in NREM sleep or anaesthesia [10–12]. It is suggested that the intrinsic loop between thalamocortical (TC) relay and RE cells plays a pivotal role in all of these behaviors by acting as a pacemaker and oscillator. The crucial mechanism at play is the rebound bursting of relay cells via hyperpolarization induced by RE inhibition.

These oscillatory behaviors primarily manifest during sleep-like brain states, where the TC cells show a prevalence for bursting [13]. Additionally, in newer studies it was shown that

thalamic integration with cortical pathways suggests a significant role of the thalamus in many higher brain functions, including sensation, attention, and cognition [14, 15].

Investigating the interaction between thalamic reticular and relay neurons at various levels is therefore crucial for deciphering the interplay of the brain with the outside world. To this end it is necessary to analyze these neuron interactions and their corresponding population activity via large-scale models. One feasible approach for scaling upwards is to employ networks of single-cell neuron models, but the computational demand rapidly increases as the network size is taken to the scale of anatomical subdivisions of the brain. For larger scales and even whole-brain simulations, it is necessary to decrease computational complexity. This can be achieved by reducing the degrees of freedom and describing homogeneous populations of neurons as the smallest units. A viable option is to use a mean-field theory to model population dynamic statistics.

Most existing neuronal field models can be separated in two groups: either phenomenological models (e.g. [16–18]), or more abstract mathematical models (e.g. [19–21]). Phenomenological models replicate biological behavior and are capable of modeling particular brain regions, cell types or whole brain recordings. However, these can not couple significant effects or characteristics to model parameters which makes it impossible to use such models far of the fitting point and renders analytical analysis impractical. Conversely, abstract mathematical models couple the dynamical aspects of neuronal activity directly to model parameters and allow analytical or fast-forward numerical analysis, but model parameters are often not well linked to biological observables.

To strike a good balance between these models, we develop in this paper a *biologically realistic* mean-field model of the thalamus that also allows analytical analysis. To achieve this biological realism with a firing rate model, our formalism follows a bottom-up approach, starting at the single-cell level and incorporating cellular and structural specificities of the thalamic circuits [22]. Our approach incorporates three crucial biological features: (1) *Irregular spiking* activity of neurons is believed to be important for transfer efficiency [23] and the correct baseline for neurons in both awake-like asynchronous (AI) states [24] as well as in sleep-like synchronous (SI) states [25]. (2) Synaptic *conductances* allow for realistic bi-stability and self-sustained activity [26] as well as modeling the fluctuation-driven regime [27]. (3) *Adaptation* mechanisms are the main generators of the different firing behaviors in the brain and important to include into models for generating realistic firing rate saturation and especially the bursting behavior of thalamic cells.

Using our mean-field model we investigate the state-dependent responsiveness of the thalamus, integrating the interplay between multiple scales (from single-cell level to the mesoscale). Building upon existing single-cell experiments we show that: First, the transition from tonic to burst firing of TC cells via ACh renders thalamic response nonlinear in sleep state. Second, sensory stimuli generate a linear response, while cortical inputs generate a nonlinear response of the thalamus. Third, cortical input and synaptic noise modulate thalamic response and synaptic noise diffuses thalamic state transitions and removes thalamic response dependency on both voltage and frequency. Finally, we demonstrate that the proposed model is capable of generating self-sustained spindle oscillations, drastically altering responsiveness in this state.

## Materials and methods

In this section we describe the single-cell, network, and the mean-field model. The chosen network and connectivity structure as well as cell and synaptic parameters are described.

## Spiking neuron model

For both single-cell and network simulations we employ the *Adaptive exponential integrate and fire model* (AdEx) (as defined in [28] and analyzed in [29]). This conductance based model often proved to be a good balance between computability and biological realism in terms of capturing all firing modes observable in real cells [30] and significantly in thalamo-cortical cells [31, 32]. Importantly, it allows for a systematic fit of real cell traces. The dynamical system is the two equations describing membrane potential $v$ and adaptation current $\omega$ of a given cell $\mu$

$$c_m \frac{\mathrm{d}v_\mu}{\mathrm{d}t} = g_L(E_L - v_\mu) + g_L \Delta\, e^{\frac{v_\mu - V_t}{\Delta}} - \omega_\mu + I_{\mathrm{syn}}(t) \tag{1}$$

$$\frac{\mathrm{d}\omega_\mu}{\mathrm{d}t} = -\frac{\omega_\mu}{\tau_\omega} + b \sum_{t_s} \delta(t - t_s) + a(v_\mu - E_L), \tag{2}$$

with the cell parameters listed in Table 1 and where $I_{\mathrm{syn}}$ models all incoming synaptic currents. It consists of two currents dependent on excitatory $G_{\mathrm{syn}}^e$ and inhibitory $G_{\mathrm{syn}}^i$ membrane conductances and is defined as

$$I_{\mathrm{syn}}(t) = (E_e - v_\mu)G_{\mathrm{syn}}^e(t) + (E_i - v_\mu)G_{\mathrm{syn}}^i(t) \tag{3}$$

$$G_{\mathrm{syn}}^{(e,i)}(t) = Q_{(e,i)} \sum_{t_s^{(e,i)}} \theta(t - t_s^{(e,i)})\, e^{-\frac{t - t_s^{(e,i)}}{\tau_{(e,i)}}}, \tag{4}$$

where $G_{\mathrm{syn}}$ is modeled such that each time a spike ($t_s$) arrives these conductances experience an increment $Q$ and exponentially relax again with time constant $\tau$. As a baseline we use $Q_e = 1\mathrm{nS}$ and $Q_i = 5\mathrm{nS}$ [22].

Additional to the integration of this ODE set comes the usual spike mechanism employed in integrate and fire models: A spike of neuron $\mu$ is counted if $v_\mu > V_{\mathrm{thr}} = -20\mathrm{mV}$, then the

**Table 1. Cell and synaptic parameters for TC and RE cells in awake (ACh) and sleep (no ACh) states.** Connection parameters see Fig 1A. The last two parameters are for the spiking network only and "-" means the same value as in awake state.

| PARAM | AWAKE | | SLEEP | | DESCRIPTION |
|---|---|---|---|---|---|
| | TC | RE | TC | RE | |
| $Q_e$ | 1nS | 4nS | - | - | exc. quant. conductance incr. |
| $Q_i$ | 6nS | 1nS | - | - | inh. quant. conductance incr. |
| $c_m$ | 160pF | 200pF | - | - | membrane capacitance |
| $E_L$ | −65mV | −75mV | −70mV | −85mV | resting (leakage) potential |
| $g_L$ | 10nS | 10nS | 9.5nS | 13nS | leak conductance |
| $\tau_w$ | 200ms | 200ms | 270ms | 230ms | adaptation time const. |
| $a$ | 0nS | 8nS | 24nS | 28nS | membrane potential adaptation |
| $b$ | 10pA | 10pA | 200pA | 20pA | spike frequency adaptation |
| $\tau_{(e,\,i)}$ | 5ms | 5ms | - | - | syn. time constants |
| $E_e$ | 0mV | 0mV | - | - | exc. reversal potential |
| $E_i$ | −80mV | −80mV | - | - | inh. reversal potential |
| $V_t$ | −50mV | −45mV | - | - | threshold for the spike onset |
| $\Delta$ | 4.5mV | 2.5mV | - | - | amplitude of the spike onset |

membrane potential is reset to $V_r = \{-55\text{mV}$ for RE, $-50\text{mV}$ for TC$\}$ for a refractory period of 5ms.

## Network architecture and model parameters

We model one thalamocortical relay (TC) and one connected reticular (RE) population of a generic lateral thalamic nucleus. We neglect interneurons, as it can be assumed that they only yield minor contribution to population dynamics [33]. One of the main potential application of the thalamus mean-field is to be incorporated into large or whole brain models with already developed cortical and sub-cortical mean-field models and related implementations [22, 34–40]. As a reference, these previous works on cortical circuits describe typically populations of $\sim 10^4$ neurons, corresponding to the size of a single cortical column. To keep the scale difference between cortex and thalamus proportional, we employ a scale of 1/10 [4, 5, 41] and therefore use $N = 500$ neurons per population. This allows to build a basic realistic-scale thalamo-cortical loop with just two mean-field models.

The network with its connections is depicted in Fig 1A. We consider a random connected *Erdos-Renyi* network comparable to the statistical assumptions of the mean-field (Table A in S1 Appendix). TC and RE populations form a loop of excitation and inhibition. TC cells do not excite other TC cells but RE cells (next to outgoing axons to the cortex). In contrast, RE are connected in an inhibitory loop and also inhibit TC cells. We propose two external drives serving as inputs to the model: The *cortical* drive $P$ (going to both populations) and the *sensory* drive $S$ (going only to TC cells) modeling cortical signals and sensory stimuli to the thalamus, respectively.

For the synaptic and connection parameter values, we start with a connection probability between TC and RE populations of $p = 5\%$, which captures the sparse connectivity between the two populations [3]. To model the dense net of locally self-inhibiting RE neurons in the

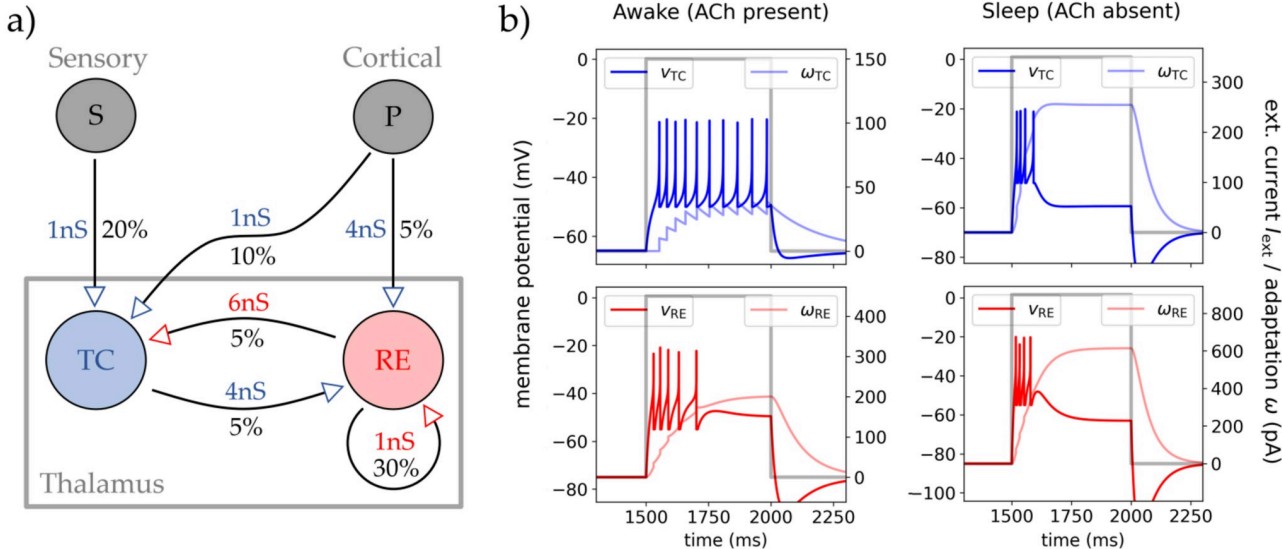

**Fig 1. Network structure and single cell dynamics. A** The chosen network structure of two thalamic populations (TC and RE cells of each $N = 500$), their synaptic increments $Q$, and connection probabilities $p$ for all connection between the TC and RE population. The external inputs are shown in gray: The *cortical* drive $P$ ($N = 8000$) and the *sensory* stimulus drive $S$ ($N = 500$). The arrows mark the direction of synaptic transmission and if they act excitatory (blue) or inhibitory (red). **B** Single cell traces from AdEx IF neurons (see Methods for details) of TC and RE for a timed constant input current (gray line). The left column shows TC and RE response to the injected current in awake state (with ACh) and the right column the same in sleep state (no ACh). The cells membrane potential $v$ and adaptation current $\omega$ are shown in color for TC (blue) and RE cell (red), respectively.

TRN [2, 43, 44], we use $p$ = 30%. There are 2 to 10 times more axons projecting from cortex to TC than from cortex to RE cells, but the amplitude of connection to RE is stronger, keeping a strong inhibitory cortico-thalamic modulation via the TRN [4, 7]. Last, the number and convergence of axons from RE to TC cells ensures sparse but strong inhibition [3, 7]. See Fig 1A for all the parameter values.

Moving to cell parameters, we model two states of the thalamus corresponding to high or low levels of the excitatory modulator acetylcholine (ACh). In [9] and [44], it was shown that low levels of ACh change the firing patterns of TC cells to inhibit single tonic firing and to promote bursting. Because of the capability of ACh to act as a switch between tonic and bursting mode in the TC cells relay, and its role in controlling the overall physiological brain state [33, 45, 46], we define here these two states as *awake* state (ACh present; wakefulness, REM sleep) and *sleep* state (ACh absent; NREM sleep, low attention).

Parameters of the single-cell model were determined via Mean-Absolute-Error (MAE) analysis between model prediction and recorded TC and RE cell traces of those two studies [9, 42] for the two states of ACh present (awake) and ACh absent (sleep). Initial parameter values are taken from Destexhe [31] with taking into account experimental parameter ranges of [9, 42] and the NeuroElectro database [52] for TC and RE neurons. The resulting parameters are shown in Table 1. The robustness of the model parameters is validated via a large range exploration in the parameter space, shown in Fig D–G in S1 Appendix Beyond this initial determination, two further modifications to the parameters values were performed: (1) A hyperpolarised $E_L$ for RE cells in sleep state. This choice does deviate from a best fit as is evident from the increased error as shown in the second row in Fig F in S1 Appendix However, this was necessary to guarantee biologically realistic stable and balanced AI dynamics of the full network (see Fig 2C). And (2) in a stronger spike adaptation $b$ for TC cells, also in sleep state. This choice is required for TC cells to burst also in network simulations (see Fig 3B) and leads to stronger bursting at the single-cell level, but does not significantly increase the error of the single-cell fit.

To show that the cells inherit the correct behaviour, using the AdEx Eq (1) with the proposed cell parameters, in Fig 1B four exemplary single cell traces of RE and TC are shown. In there a constant–time gated–current was injected in to the cell to invoke a firing response of the cell. This was done by setting a rectangular pulse as $I_{syn}$ in Eq (1) ($I_{syn}$ generates tonic and burst firing via two different bifurcations depending on excitability state, see [29]). The top row shows the wanted response types for the TC cell: Tonic firing with awake parameters (modulating ACh), and burst firing with sleep parameters (low-level of ACh). In the bottom row, RE cell's respond via burst firing in both parameter states, but the burst duration decreases in sleep state while keeping the same amount of spikes (increased *burstiness*).

## Mean-field model

El Boustani & Destexhe [47] developed a second-order mean-field formalism of differential equations describing the firing rate statistical moments of spiking networks. This general framework closes the statistical hierarchy at second order and is applicable to any arbitrary neuron models as long as a characteristic transfer function can be defined. It is assumed that the network is a sparse and randomly connected *Erdos-Renyi model*. It is derived with the assumption of the system being in an E-I balanced AI state. This formalism is extended by including the slow dynamic effects of adaptation [22] so that the system is fully described by mean firing rate $\nu_\mu$ and adaptation $\omega_\mu$ for each neuron population $\mu$. The differential equation

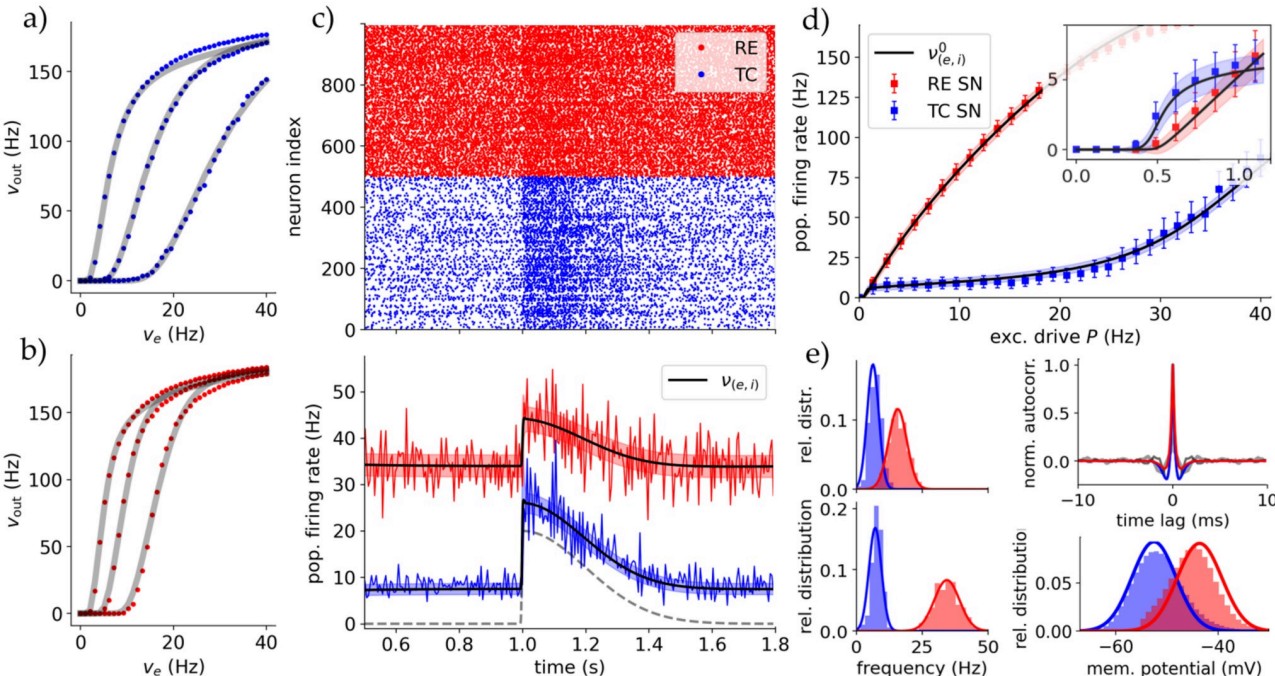

**Fig 2. Validating the mean-field with spiking networks. AB** The fitted transfer functions for RE and TC cell for three different inhibitory inputs each with their corresponding single cell simulations. Top (A, blue) is for TC and bottom (B, red) for RE cell-type (in awake state). The dots each represent the averaged firing rate of a cell over 100 runs. **C** Comparison of the firing rate of the mean-field and the spiking network for constant cortical drive $P = 4$Hz and a split-Gaussian stimulus coming from $S$. Top is the raster plot showing all spiking times $\{t_s\}$ for all neurons in the spiking network simulation. Bottom is the averaged mean firing rate of spiking network (blue/red lines) and predicted mean firing rate of the mean-field $\nu$ (black line) with its standard deviation (shaded blue/red areas). **D** Comparison of the equilibrium firing rate of the spiking network and of the mean-field over a range of cortical inputs. Each dot represents a spiking network simulation for 10s where the steady long time mean is calculated. The black lines correspond to the mean-fields fixpoints $\nu_{(e,i)}^0$, with the shaded areas being the standard deviations. The *inhibited* regime between ca. $P \simeq (1, 20)$Hz marks the standard activity employed. The inset shows a zoom at the low-drive regimes where activity is first silent and then controlled by TC until $P \simeq 1$Hz. **E** Left column: The firing rate distributions of spiking network (histogram) and mean-field (line) for $P = \{2, 4\}$Hz. Bottom-right: Comparison of membrane potential distribution for 4Hz. Top-right: Autocorrelation $\tau_{ac}$ of TC and RE population for spiking network (grey lines) and mean-field (blue/red lines).

system for this framework then reads

$$T\frac{\partial v_\mu}{\partial t} = (F_\mu - v_\mu) + \frac{1}{2}\partial_\lambda\partial_\eta F_\mu c_{\lambda\eta} \tag{5}$$

$$T\frac{\partial_t c_{\mu\nu}}{\partial t} = \delta_{\mu\nu}A_{\mu\mu}^{-1} + (F_\mu - v_\mu)(F_\nu - v_\nu) + \partial_\lambda F_\mu c_{\nu\lambda} + \partial_\lambda F_\nu c_{\mu\lambda} - 2c_{\mu\nu} \tag{6}$$

$$\partial_t\omega_\mu = -\frac{\omega_\mu}{\tau_\omega} + bv_\mu + a((\mu_V)_\mu - E_L), \tag{7}$$

where $F_\mu$ is the transfer function of cell population $\mu$ and $c_{\mu\nu}$ the covariance between two population's firing activity. The indices $\{\mu, \nu, \lambda, \eta\}$ run over the set of populations, e.g. in our case of two populations the set of $\{e, i\}$ for excitatory TC and inhibitory RE. The derivatives are defined as $\partial_\mu = \frac{\partial}{\partial v_\mu}$. Important to note is the role of $T$ which marks the adiabatic time step such that dynamics with smaller time resolutions are not captured and which has to fulfill the requirements listed in Table A in S1 Appendix

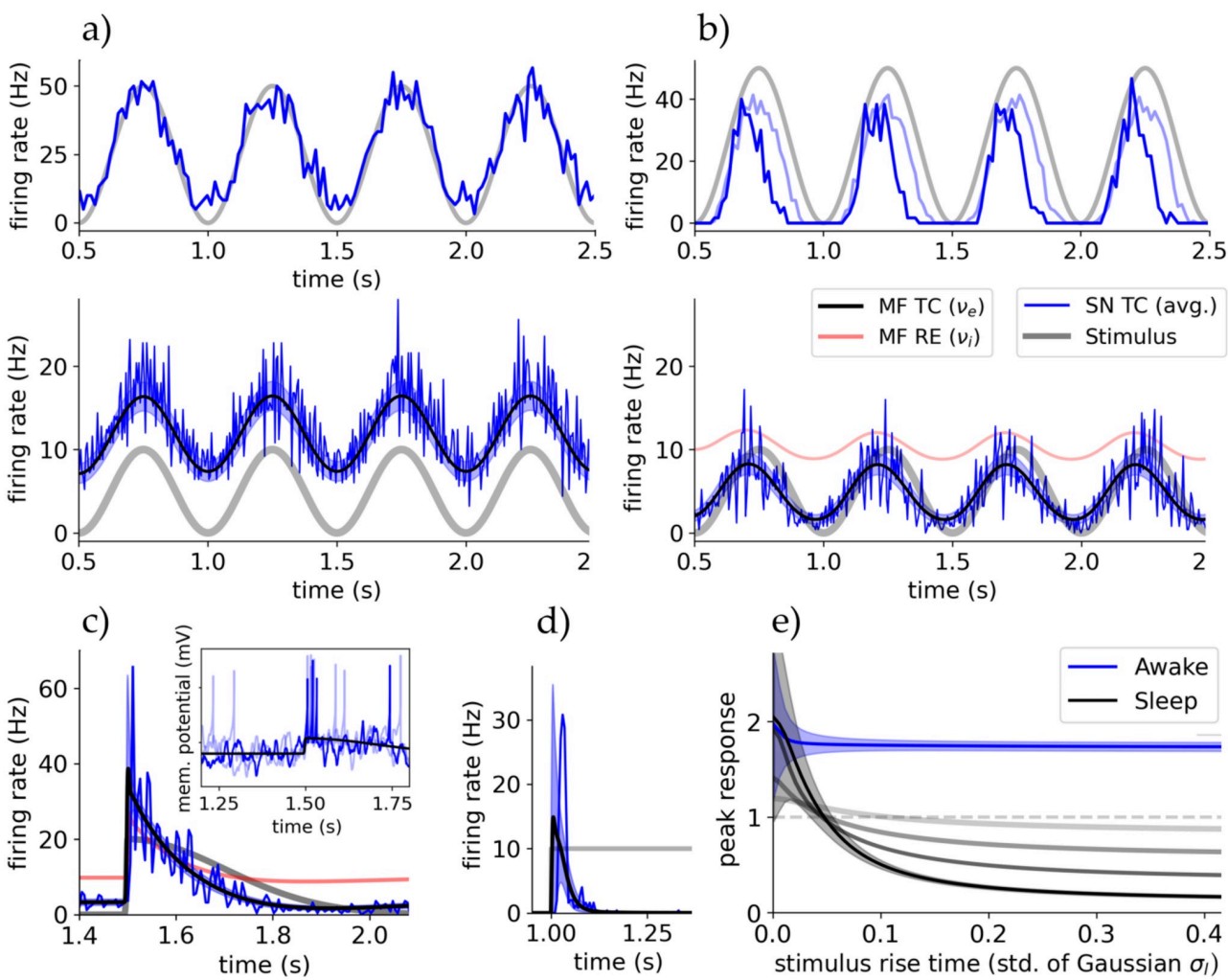

**Fig 3. Bursting of TC cells renders thalamic response nonlinear in sleep state. A** Top row: Single cell and population response to a strong oscillatory sensory drive $S$ in awake state. Bottom row: Activities of spiking network (TC population, blue) and mean-field (TC population, black line with color-shaded std., RE population, red line). The grading stimulus is pictured in light grey. The single cell recording of the top rows is taken from this network simulation. RE activity is of same frequency and phase as TC activity but with amplitude $\sim 50$Hz. **B** The same setup as in A but in sleep state (TC bursting, see main text). The single cell recording is done in the network of A which was in awake state to be close to the experiment. (Dark blue is sleep state and light blue is sleep state with lower adaptation $b = 20$pA.) The single cell traces in A and B reproduce the experiments of [7]. **C** Thalamic response of spiking network and mean-field to a fast changing stimulus (split-Gaussian with steep left-hand std. $\sigma_l$). Inset shows trace of 3 random TC cells of the spiking simulation, showing bursting at the onset of the stimulus ($t_0 = 1.5$s; mV per s). **D** Rectangular stimulus in absence of cortical drive showing that TC activity vanishes after initial burst. **E** The maximum amplitude of response (peak), relative to the incoming stimulus amplitude, as a function of the 'slope' of stimulus (left std. $\sigma_l$ of split-Gaussian; amplitude 10Hz and right std. 0.2s). Depicted is the response for awake and sleep state and in sleep state for different applied cortical constant drives (from black to gray: {1, 2, 4, 10}Hz).

The core of this formalism is the transfer function $F$ and so the main task in constructing a mean-field of the thalamus is to get the transfer function of TC and RE cells in the two states of awake and sleep.

To derive the transfer function we follow the semi-analytical approach of Zerlaut et al. [48] which combines the seminal studies of [49, 50]. In there the firing rate is written as a probabilistic function counting the spikes in term of the membrane potential $v(t)$ being above a certain *spike threshold potential* $V_\theta$ in each time bin of duration $\tau_V$ which resembles the membrane potentials autocorrelation time. In the Gaussian limit we get a function dependent on the

membrane subthreshold fluctuation statistical moments and define that as our transfer function

$$F(\mu_V, \sigma_V, \tau_V) \equiv v_{\text{out}} = \frac{p(v > V_\theta)}{\tau_V} \overset{\text{Gaussian}}{=} \frac{1}{2\tau_V} \text{erfc}\left(\frac{V_\theta^{\text{eff}} - \mu_V}{\sqrt{2}\sigma_V}\right),$$ (8)

where $\mu_v$ is the mean and $\sigma_v$ the standard deviation of the (subthreshold) membrane potential. In the second step, the constant threshold $V_\theta$ is replaced with a phenomenological one acting as a function dependent on—and therefore accounting for—different cell properties. Because there is no theoretical form, a general second order polynomial dependent on the set $\{\mu_V, \sigma_V, \tau_V\}$ was proposed [49]

$$V_\theta^{\text{eff}}(\mu_V, \sigma_V, \tau_V^N) = P_0 + \sum_x P_x \cdot \frac{x - x^0}{\delta x^0} + \sum_{x,y} P_{xy} \cdot \frac{x - x^0}{\delta x^0}\frac{y - y^0}{\delta y^0},$$ (9)

with $x, y \in \{\mu_V, \sigma_V, \tau_V^N\}$ and where $\tau_V^N = \tau_V \frac{g_L}{c_m}$ is the non-dimensionalised autocorrelation and the parameters space is normalised to limit the fluctuation driven regime, with mean $x^0$ and deviation $\delta x^0$.

Here either single cell simulations or experimental clamp data can be used to get values for the unknown amplitudes $\{P\}$. This fitting has to be performed for each distinctive cell type, so in our case for TC and RE neurons. Because the two states awake and sleep are mostly changes in adaptation parameters, and it was shown in [22] that the mean-field is predictive even far from its fitting point, we just need one fit per cell type. This also is biologically realistic, for the changes induced by e.g. ACh would not change the cell morphology, and we consider the threshold membrane potential to stay the same for both states.

The set of $\{\mu_V, \sigma_V, \tau_V\}$ can be calculated purely analytically by using Campbell's theorem and assuming Poissonian distribution of incoming spikes as the generator of subthreshold fluctuations [50] (as is the case in the AI regime). The mean or static synaptic conductances are calculated then as a function of incoming spike frequencies $\{v_e, v_i\}$ in terms of their mean and standard deviation

$$\mu_{G(e,i)} = v_{(e,i)} K_{(e,i)} \tau_{(e,i)} Q_{(e,i)}$$ (10)

$$\sigma_{G(e,i)} = \sqrt{\frac{v_{(e,i)} K_{(e,i)} \tau_{(e,i)}}{2}} Q_{(e,i)},$$ (11)

where $K_\mu = p_\mu N_\mu$. With that the general input conductance of the cell can be computed

$$\mu_G(v_e, v_i) = \mu_{Ge} + \mu_{Gi} + g_L.$$ (12)

Then we can calculate the mean membrane potential $\mu_V$ from the first order approximation of Eq (1) as a function of incoming spike frequencies

$$\mu_V(v_e, v_i, \omega) = \frac{\mu_{Ge}E_e + \mu_{Gi}E_i + g_L E_L - \omega}{\mu_G}.$$ (13)

Taking Eq (3) as the general synaptic input, we can calculate the form of a single postsynaptic potential (PSP). And via shotnoise theory get the density power spectrum of membrane fluctuations $P_V(q)$ as a response to a stimulation Eq (3). Then the variance of fluctuations with

taking the integral in frequency domain $\sigma_V^2 = \int_q P_V(q)$, follows to

$$\sigma_V(v_e, v_i) = \sqrt{\sum_{(e,i)} K_{(e,i)} v_{(e,i)} \frac{(U_{(e,i)} \cdot \tau_{(e,i)})^2}{2(\tau_m + \tau_{(e,i)})}}, \tag{14}$$

where $\tau_m(v_e, v_i) = \frac{C_m}{\mu_G}$ and $U_{(e,i)} = \frac{Q_{(e,i)}}{\mu_G}(E_{(e,i)} - \mu_V)$ is the effective synaptic drive.

Finally, the autocorrelation time $\tau_V$ completes the framework which is defined in terms of the power spectrum as $\frac{1}{2}\frac{P_V(0)}{\int_q P_V(q)}$, resulting in

$$\tau_V(v_e, v_i) = \sum_{(e,i)} K_{(e,i)} v_{(e,i)} (U_{(e,i)} \cdot \tau_{(e,i)})^2 \cdot \sum_{(e,i)} \frac{\tau_m + \tau_{(e,i)}}{K_{(e,i)} v_{(e,i)} (U_{(e,i)} \cdot \tau_{(e,i)})^2}, \tag{15}$$

where in case of only one synaptic event this would reduce to $\tau_V = \tau_m + \tau_{(e,i)}$.

With Eqs (13), (14) and (15) the transfer function with effective threshold Eq (8) is now dependent only on the incoming firing rates at excitatory and inhibitory synapses $F(\mu_V, \sigma_V, \tau_V) \rightarrow F(v_e, v_i)$, closing our firing-rate based mean-field formalism.

## Transfer function fit

To get the transfer functions we fit $\{P\}$ on single cell simulations of TC and RE cells (in awake state) using the AdEx Eq (1). The formalism translates excitatory and inhibitory input firing rates $\{v_e, v_i\}$ of a neuron into its fluctuation statistics $\{\mu_V, \sigma_V, \tau_V\}$ and then to its output firing rate.

The advantage of this semi-analytic approach is that–given either simulated or experimental data–we can calculate the phenomenological threshold $V_{\text{thr}}^{\text{eff}}$ via reordering of Eq (8). Then the employed procedure is to first fit Eq (9) linearly in the threshold space (depending on the topography of the space to capture, this fit can be done nonlinearly too). However, here Eq (13) has to be adjusted because the adaptation $\omega$ is unknown. Therefore, the (stationary) solution to Eq (7) will be used to calculate $\omega$ from the firing rate data. The resulting values for $\{P\}$ are following used as initial guesses for the fully nonlinear fit of Eq (8) in the original firing rate space.

For the fit we normalised the fluctuation regime the same way as done in previous works [22, 49]; to ensure comparability: $\mu_V^0 = -60\text{mV}$, $\delta\mu_V^0 = 10\text{mV}$, $\sigma_V^0 = 4\text{mV}$, $\delta\sigma_V^0 = 6\text{mV}$ and $\tau_V^{N\,0} = 0.5$, $\delta\tau_V^{N\,0} = 1$.

## Results

The results of this paper are structured as follow: First the mean-field model will be compared with simulated spiking network dynamics and validated in and far from the fitting point. Then thalamic responsiveness and how it depends on different external and internal states will be investigated. Lastly, spindle oscillations in a sleep-like state in the employed models are shown.

### Fitting and validation

In this section, we validate the mean-field model and demonstrate its suitability for modeling both awake and sleep state of the thalamus by comparing it with spiking networks.

The fit parameters of the mean-field's transfer function via Eq (9), obtained using our fitting technique, are depicted in Table 2. These parameters are applied to both awake and sleep states (ACh absent/present) and are used throughout this and the following three sections.

**Table 2. The fitting parameters of $V_{\text{thr}}^{\text{eff}}$ and their values.** All values in mV.

| cell | $P_0$ | $P_\mu$ | $P_\sigma$ | $P_\tau$ | $P_{\mu\mu}$ | $P_{\mu\sigma}$ | $P_{\mu\tau}$ | $P_{\sigma\sigma}$ | $P_{\sigma\tau}$ | $P_{\tau\tau}$ |
|------|-------|---------|-----------|----------|--------------|-----------------|---------------|--------------------|------------------|----------------|
| TC | −47.31 | 1.68 | 0.97 | −3.46 | 0.47 | −1.68 | −6.46 | 3.43 | −1.14 | 0.19 |
| RE | −40.77 | −1.98 | −3.12 | 3.57 | 1.39 | −0.38 | −0.33 | 0.16 | 0.26 | −0.53 |

In Fig 2A, we show the fitted transfer functions *F* for TC cells (top, blue) and RE cells (bottom, red) across the full range of excitatory input frequencies ($\nu_e$) and a subset of three inhibitory input frequencies ($\nu_i$). Each dot represents the averaged output frequency from the single-cell simulations over 5 seconds. The sigmoid shape of the transfer function (8) is evident. Certain deviations from the fitted predictions via *F* are observed only at very high firing rates; a region of lesser biological relevance for the phenomena studied in this paper. To improve statistics, the single-cell firing rates were averaged over 100 runs.

A direct validation of the mean-field model is to compare the predicted mean firing rates and their standard deviations with those of the full spiking network, both modeling the entire thalamic substructure (Fig 1A). This comparison is shown in Fig 2C. Both populations receive an external constant *cortical* input of *P* = 4Hz and a split-Gaussian *sensory* stimulus *S* (definition in Section B in S1 Appendix). The spiking network provides the membrane potential evolution and spiking times $t_s$ for all cells. The spikes of all neurons are shown in the upper raster plot. By averaging the number of spikes over a specific bin time $T_{\text{bin}}$, we calculate the time-dependent averaged firing rate of the spiking population. We use $T_{\text{bin}}$ = 5ms for all simulations except stated otherwise. To compare to the mean-field, in the formalism we have to employ a similar time window for the mean-fields time constant *T*, and we set $T = T_{\text{bin}}$ (in accordance with the formalism requirements, Table A in S1 Appendix). The spiking network and mean-field show the wanted balanced excitation-inhibition (E-I) state in AI regime with RE activity being dominant.

In Fig 2D, we vary the cortical drive *P* and compare the equilibrium or stationary population firing rates (see Section C in S1 Appendix) for TC and RE populations in both the spiking network and mean field over a 10-second simulation. This analysis reveals four distinct regimes of TC response: The first regime with no activity. The second regime with a fast response to changes in *P*. The *inhibited* third regime with limited responses. And the fourth regime with strong TC cell responsiveness due to (biologically unrealistic) saturated RE cell activity. This justifies using a cortical drive $1 < P < 10$Hz for most simulations, ensuring a stable low-activity AI state, comparable to *in-vivo* experiments.

In Fig 2E, we compare the distribution of firing rates and membrane potentials. In the latter the refractory states are removed to get a realistic comparison with the mean-field. The fit between mean-field and spiking network distributions only diverges at high firing rates of close to 100Hz due to the discontinuous nature of spiking models. The good agreement in not only firing rate but also membrane potential is significant, because Eqs (13) and (14) predict accurately the spiking populations membrane potential statistics and can henceforth be used to compare with electrophysiological data and methods.

In the same figure, top-right, there is depicted a comparison of (normalised) autocorrelations $\tau_{\text{ac}}$ of TC and RE population activity in the stationary state corresponding to *P* = 4Hz, showing a strong independence of population activity as expected from a inhibition-controlled network without excitatory-excitatory connections. This also agrees with the models being in AI state and the choice of $T = T_{\text{bin}} = 5\text{ms} > \tau_{\text{ac}}$ is justified.

Finally, we assess the robustness of the mean-field by varying global parameters (in Fig B in S1 Appendix). This is done for adaptation parameters {*b*, *a*}, which exhibit the significant

change between awake and sleep state (Table 1), and synaptic excitatory conductance $Q_e$ to validate its change for simulations in this study. We demonstrate that even far of the actual fitting point, the mean-field remains effective in capturing network dynamics. This validation allows us to use the mean-field approach for parameter space analysis and the study of the transition between awake and sleep states with just one mean-field parameter fit.

## Tonic and burst firing modes

We explore how bursting (the state of ACh neuromodulation) impacts the response of thalamic neurons and their network. Based on the fit to biological bursting TC cells from [42] we can already state that the employed parameter set with the AdEx shows *bursting* of single TC cells in the ACh-depleted or sleep state (as evident from the cell traces in Fig 1).

We want to investigate the stability of those regimes and their dependence on model parameters. With the employed models, the mechanism generating bursting is the slow adaptation current of the AdEx (1). In Section E in S1 Appendix we derive an analytic metric quantifying firing adaptation using the transfer function of our mean-field framework. With this metric and with single-cell scans, we show in Fig A in S1 Appendix that the awake and sleep states are well separated. While they are stable to small perturbations, they are also close to the phase transition which ensures richer dynamics.

Following, we aim to replicate experiments at single cell level on tonic and bursting states of TC cells, as documented by Sherman & Guillery [7, ch. 6]. These experiments involved manipulating the membrane potential of recorded TC cells to force either a tonic mode (around −65mV, resting state) or a bursting mode (around −75mV, hyperpolarised state), in the absence of external stimuli. A grating retinal stimulus was applied, leading to an oscillatory firing rate response. There, TC cells in tonic mode exhibited a linear response, while TC cells in bursting mode showed responses primarily during the initial phase of each stimulus period.

We recreated this behaviour computationally in our proposed spiking network with an oscillatory sensory drive *S*, with amplitude of 10Hz and frequency of 2Hz. The network was set in awake state emulating a lightly anaesthetised state as in experiment. To model the thalamus *in-vivo*, a constant external cortical drive of *P* = 4Hz was applied (the *inhibited* regime, Fig 2D). Subsequently, we recorded one single cell with each awake and sleep parameters. While the proposed awake and sleep states are not identical to the artificially set tonic and bursting modes in the experiment, the switch via acetylcholine (ACh) generates a similar polarization.

The recorded cell's response was calculated by averaging the spike times over 40 simulations for a time bin of 15s. This *firing rate* is depicted in the first row of each Fig 3A and 3B for the awake state and sleep state, respectively. We observe the same response patterns as in the experiment for awake and sleep parameters, although with slightly lower response amplitudes in the sleep state compared to the hyperpolarized state of the experiment. This can be attributed to the absence of T-channels and low-threshold spikes in the AdEx model [7]. In Fig 3B there is also depicted, in light blue, the response in sleep state with adaptation parameter in line with stated constraints (*b* = 20pA), which does not show the correct behavior.

Moving to population-level, the second row of Fig 3A and 3B superimposes the spiking network's and mean-field's responses. In the awake state the entire TC population faithfully tracks the stimulus, as do single TC cells (top row). In sleep state, the response amplitude and also RE activity are greatly reduced, both showing phase locking while keeping the shape of the stimulus. The phase shift is created by the delay of slowly activating adaptation mechanisms and reactive RE inhibition. Phase locking was found in sleep state for all amplitudes of stimuli.

The effects, however, are quite small and functionally not so different between awake and sleep states. We would expect stronger effects of bursting in the responsiveness when

adaptation effects are significantly slower than changes in the input and subsequently membrane potential (as is the case for single cells, Fig 1B for a rectangular pulse). To investigate this at the network level, thalamic response to faster changing stimuli is tested. In Fig 3C, a split-Gaussian with steep left-hand std. is depicted (at $t_0$ = 1.5s with std. $\sigma_l$ = 2ms and amplitude $A$ = 20Hz, see Section A in S1 Appendix; inhibited regime). TC response is two-fold at an initial peak and then quickly adapts. This response curve is nonlinear and does not follow the shape of the stimulus faithfully anymore. This *peak* response is a direct effect of TC cells bursting at the onset of the stimulus, as shown in the inset for a random TC cell of the spiking network simulation. Similar to the single cells definition of showing bursting (Fig 1B), also the TC population activity vanishes after the initial peak for a sustained input (no cortical drive, Fig 3D). The initial bursting of TC cells is captured by the mean-field mainly via its second order moments, namely autocovariance $c$ (blue shaded areas in plot) and autocorrelation $C$ next to a smaller increase in mean firing rate $v$.

To analyze the dependence of thalamic response for both tonic and bursting TC cells (awake and sleep state) on the shape of the stimulus, in Fig 3E, there is depicted the *peak* response amplitude of the thalamus as a function of the std. $\sigma_l$ of a split-Gaussian stimulus ($\sigma_r$ = 0.2s and $A$ = 10Hz), representing the change or 'shape' of a generic stimuli. In awake state the peak response is nearly constant, does not depend on how fast the stimulus changes, and the thalamus magnifies the input amplitude nearly two-fold. In contrast, in sleep state, only steep slopes or fast changing stimuli are generating a substantial response, whereas for slowly changing stimuli the response is drastically reduced (Fig 3B).

In conclusion, both single-cell and population-level response of TC cells appears linear in awake state (ACh present) with enhanced stimulus amplitude, while in sleep state (ACh absent) response is linear but of reduced amplitude for slowly changing stimuli, and nonlinear for quickly changing stimuli. In addition, and as evident from Fig 3C, both spiking network and mean-field model capture the bursting of TC cells, resulting in a "bursting" population response. This enhances stimulus detection in low attention states for significant sensory inputs and the transmission of mostly time-dependent information such as oscillations in sleep state.

## Cortical and sensory input

We will proceed with how thalamic responsiveness depends on background activity and how the two different biological inputs to the thalamus modulate it's behaviour.

Referring back to Fig 3D, we see a modulating role of cortical input, which in sleep state can render the usually highly nonlinear TC response linear by removing the dependence on stimulus change at high cortical drives (gray lines in plot). This could allow the cortex to generate a time window where outside information temporally is transferred faithfully during usually non-attentive states.

Moving on, we are interested in the differences between the two drives. In Fig 4A, the (stationary) firing rate response of the TC cell population in the mean-field model for different constant inputs in awake state is displayed, with both cortical and sensory drives. We applied a small constant cortical input $P$ = 1Hz to be in a low activity AI state comparable to *in-vivo* (Fig 2D). In case of sensory stimuli, the response is strongly proportional to the input, and we identify that the slope of this response is influenced by the cortical drive $P$.

In Fig 4B we see this dependency is inversely proportional, where we conducted simulations for varying cortical drive amplitudes and observed that the gain (slope of the linear response curve) decreases as $P$ increases. In the sleep state, the response remains relatively constant, slightly decreasing with $P$, contrasting the awake state's high gain for all cortical inputs.

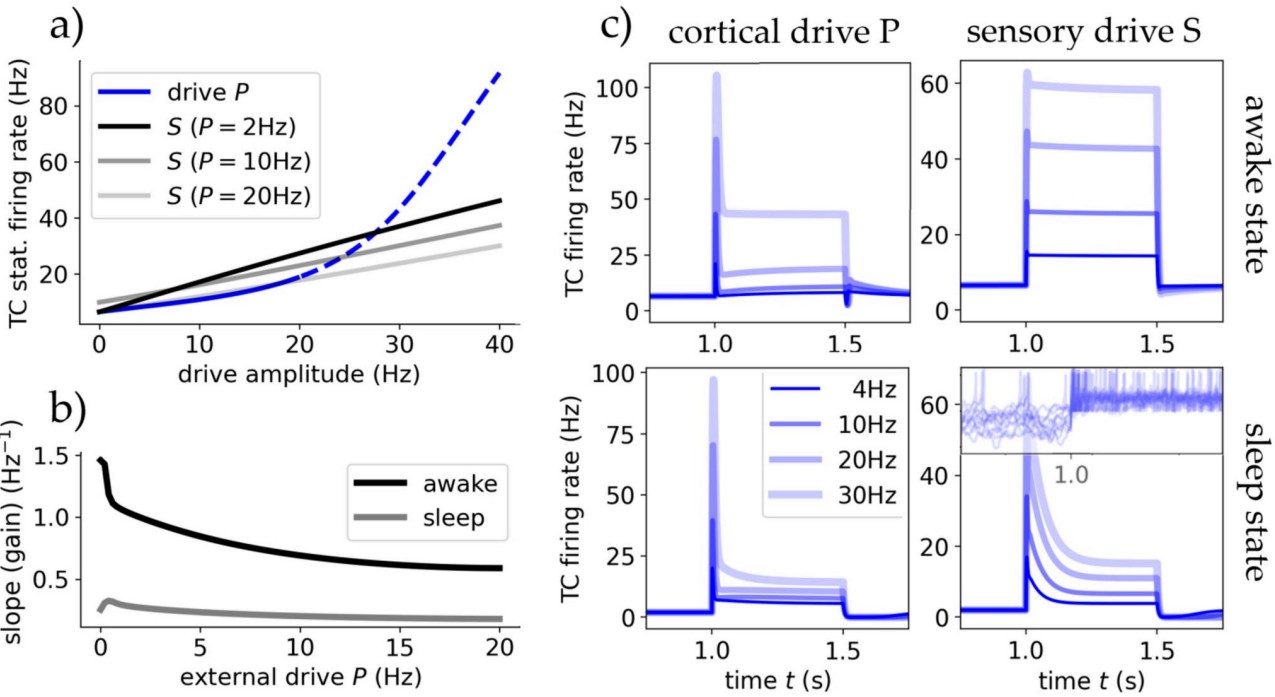

**Fig 4. The thalamus' responsiveness depends on external input origin. A** The steady state output firing rate of the TC population after reaching equilibrium for different drives P and S. Blue is for inputs coming from cortical drive *P*, where solid marks the *inhibited regime* and dashed the *blow-up* regime. Black are for sensory drive *S*, with varying degrees of cortical input. TC cells respond linearly for sensory stimuli, whereas cortical stimuli are nonlinear only showing a strong proportionality after ∼20Hz. **B** Cortical drive removes thalamic response dependency on stimulus frequency. The gradients of the sensory input response curves (black in A as a function of cortical input *P* for awake and sleep state. **C** The TC populations response to a rectangular stimulus of varying amplitude coming from either drive in both states. In the bottom-right there are also depicted 10 randomly chosen single cell traces to connect the population spike with single cell burst-like behaviour (mV per s).

The cortical drive removes the firing rate-dependency of thalamic response to stimuli in awake like states but does not alter it in sleep state. This is in agreement with studies which assumed the cortical role in the thalamus to be modulating thalamic response similar to noise [51], and with our study on synaptic noise.

For cortical input, the response is nonlinear but exhibits multiple linear regions, as seen in Fig 2C. The threshold at around 25Hz serves as a turning point (the end of the *inhibited* regime, at which the RE population firing rate saturates). Inputs below this threshold do not provoke a strong sustained response, while inputs above do. The RE population's strong response to changes in the *inhibited* regime nearly nullifies TC and therefore thalamic response.

These behaviors are evident in the TC population's response to a rectangular pulse stimulus from either *P* or *S* in Fig 4C. Notably, low cortical inputs can even be repressive, with only larger amplitudes triggering robust and sustained responses, particularly in the awake state (in agreement with studies such like [6]). In sleep state for both inputs or with low cortical inputs in awake state, responses are highly nonlinear, emphasizing the transfer of gradients rather than absolute values. The initial activity spikes at the onset of the input are created by the delay it takes the RE population to react to both stimulus and TC excitation to inhibit TC activity and –to a lesser extent– by the delayed adaptation mechanisms of both RE and TC populations. This is magnified in sleep state by stronger adaptation effects and resulting single cell bursting. This mechanism allows the thalamus to respond to cortical input and modulation despite its strong inhibiting effect via the TRN.

Concluding, only in awake state and for sensory input, or with cortical control for sensory input at sleep state, thalamic responsiveness is linear while only temporal information is transferred for cortical input and sensory input at sleep states without cortical control.

## Synaptic noise

We have analyzed so far how the responsiveness of the thalamic cells depends on the different firing modes and input sources. However, it has been shown that the level of synaptic noise (background activity) can significantly change these responses. We analyse in this section the role of noise as background synaptic and subthreshold activity and how it influences response and firing modes. We start by replicating single cell findings from Wolfart [53]. They observed that synaptic noise controls TC neurons response and behaviour and that such noise removes the dependency of TC cells response on voltage and input frequency.

We recreated this computationally at single cell level. Fig 5A shows the response of single TC cells in awake state to a Poissonian spike train of 5Hz with varying excitatory synaptic strength ($Q_e$), reflecting the experimental setup. We observe the same step-like function in the static case without external synaptic noise: going from no activity to single spike response to double spike response or bursts at high conductances (regions separated by dashed lines). With noise the response function becomes smoother and the partition of the aforementioned regimes becomes blurred. The time-dependent noise was implemented as an Ornstein-Uhlenbeck (OU) process entering the cells membrane potential as a synaptic current (see Section B in S1 Appendix).

To translate this behaviour to the population level we did simulations of the full spiking network of the employed thalamic substructure. A constant Poissonian input of 15Hz was inserted into all cells, coming from just one source; comparable to dynamical patch clamps at single cell level. The stationary firing rate output of the TC population was measured for different synaptic strengths $Q_e$. The resulting response function is depicted in Fig 5B for the static and noisy case for both spiking network and mean-field.

For the mean-field, the noise-dependent shape of the response function is passively included in the definition of the transfer function (8), with its slope being controlled by the standard deviation of the subthreshold membrane potential ($\sigma_V$). However, to recreate the experiment, which employed a time-dependent external noise, we extended the formalism by adding two additional static synaptic conductances $\tilde{\mu}_{G(e,i)}$. Those are modelled as OU-type functions averaged for each time bin equal to the mean-field's time constant $T$ (see Section B in S1 Appendix).

Both spiking network and mean-field show that the TC populations response function has its maximum slope at the same place as the first step at single cell level from no activity to single spike response (the first dashed line in Fig 5A and the dashed line in Fig 5B, respectively). Furthermore, the effect of synaptic noise is the same for population response as in the single-cell experiment, decreasing the response functions maximum slope (see Fig 1C in [53]).

How the maximum slope of the response function depends on this noise is depicted in Fig 5C (compare to single-cell experiment; Fig 1c inset in [53]). Here instead of the injected noise the noise-dependent membrane potential subthreshold fluctuations averaged over all runs ($\bar{\sigma}_V$) is shown. In sleep state the population response slope is $\sim 20\%$ less steep for the static case or small noise. Strong synaptic noise and subsequent membrane potential fluctuations decrease the slope as expected. Additionally, synaptic noise diffuses the response differences of awake and sleep state at intermediate noise levels and removes nearly all dependence of thalamic response on conductance at high noise levels ($\bar{\sigma}_V > 10$mV), where the response function

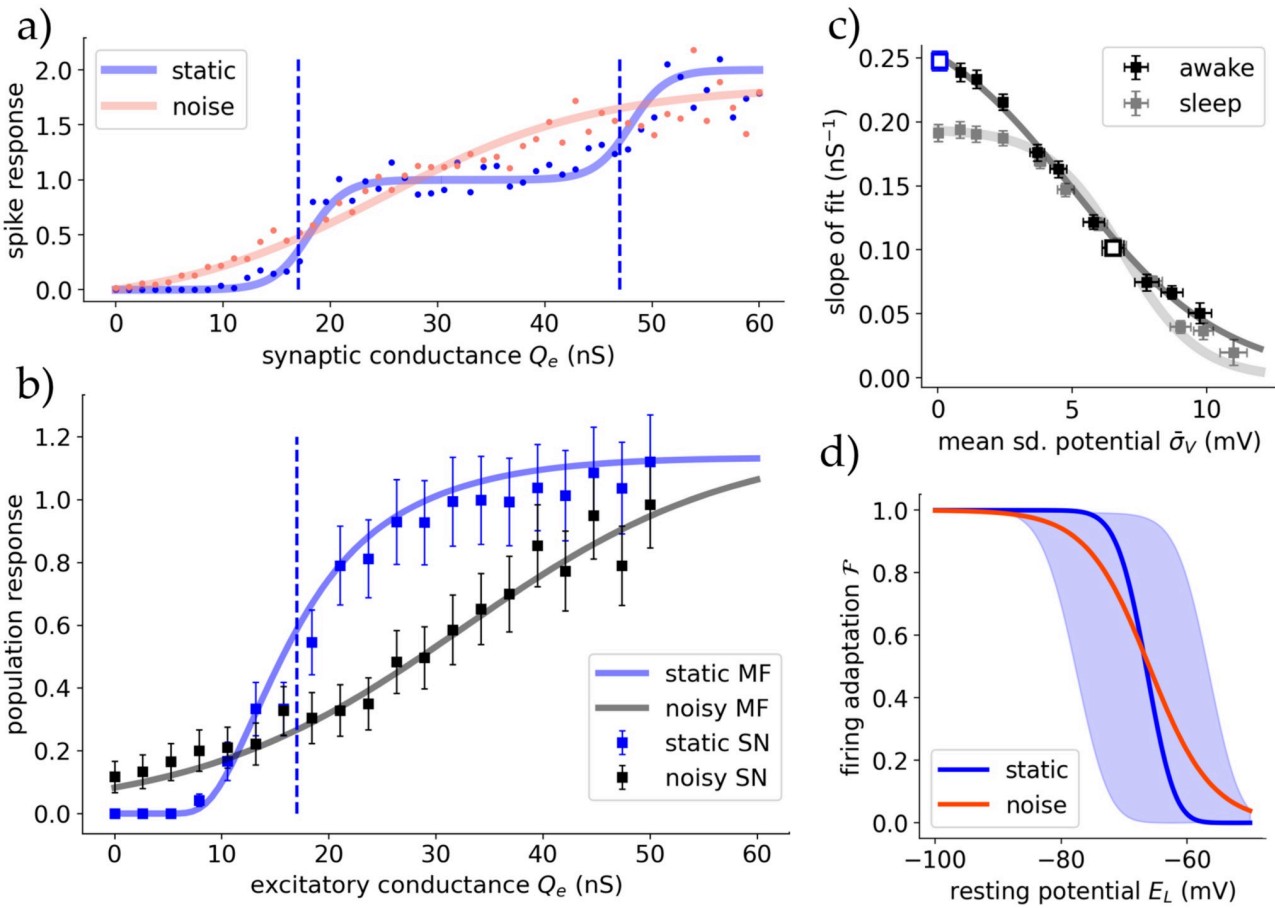

**Fig 5. Synaptic noise modulates the dependence of thalamic responsiveness on voltage. A** Response to a 5Hz Poissonian spike train for different values of excitatory synaptic strength $Q_e$ for simulated spiking single cells in awake state. The noise was injected as an OU-like current in the membrane potential via $I_{syn}$. The dots represent each the spikes per receiving incoming spike, averaged over 100 runs for 10s each. The lines correspond to sigmoidial fits, where the blue dashed lines mark the shift fit parameter depicting the center of the slope. Reproducing Fig 5A of [53]. **B** The same setup but with the full thalamic spiking network (squares) and the mean field (lines), showing the relative response to a 10Hz Poissonian spike train. The dotted line marks the slope center of the single cell simulations going from no spikes to a one-to-one spike response. For the mean-field the synaptic noise was added as an additional time-dependent conductance into the formalism (see main text). **C** The (maximum) slopes of the mean-field response curves A plotted against the standard deviation of the membrane potential predicted by the mean-field. Showing a proportionality between fluctuations and the slope of the response function. At high noise levels the difference between awake and sleep state vanishes. The two cases from B are drawn as empty blue/black boxes. **D** Synaptic noise reduces the dependency of TC firing adaptation (sim. burstiness) on cell state (polarisation). Shaded area is the standard deviation induced by small conductance noise (5nS), and orange the average.

is nearly constant (at a value dependent on the ratio of excitatory and inhibitory noise $\tilde{\mu}_{Ge}/\tilde{\mu}_{Gi}$).

Additionally, the effect of synaptic noise on the firing adaptation $\mathcal{F}$ of TC cells was tested in Fig 5D. Noise diffuses the state transition between no firing adaptation and strong firing adaptation for different levels of membrane potential polarization. As before we can refer to the similarity of $\mathcal{F}$ to *burstiness*, and hypothesize that strong noise allows for firing adaptation and also bursting for membrane potential levels showing no bursting without noise. Although the effect for the noise studied here is smaller, this is qualitatively in agreement with experimental study ([53] Fig 5b therein).

Previously, we showed that synaptic noise modifies thalamic response dependency on voltage and conductance. There, input frequency was fixed. Further following [53], we proceed to investigate how noise changes the thalamus' response in respect to input frequency.

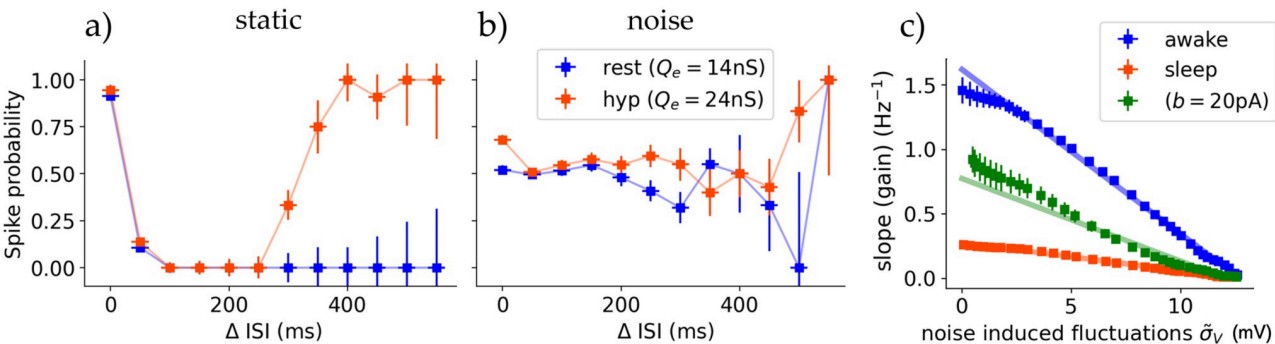

**Fig 6. Synaptic noise removes the dependence of thalamic response on frequency. A** A single TC cell's spike response probability dependent on the interspike intervall (Δ ISI) between input spikes. The input is a Poissonian spike train with a mean frequency of 10Hz, comparable to an in-vivo retinal input. For both resting ($E_L$ = −65mV, blue curve) and hyperpolarized ($E_L$ = −70mV, orange curve) states the spike response is nearly 100% at low ISI's and therefore only reacting to *summed* input spikes. In hyperpolarized state, with T-channel adjusted synaptic conductance (see main text), the TC cell responds to also high ISI's with a nearly one-to-one spike probability. This reproduces Fig 4 in [53]. **B** Same setup as in A but with additional synaptic noise (see main text). Frequency-dependent response is nearly removed. **C** Thalamic stationary response slope (gain per increase of input, see main text) of the mean-field to gated sensory stimuli of 10Hz as a function of synaptic noise via noise induced membrane potential fluctuations. For the awake state, and the sleep state with $b \in \{20, 200\}$pA. Regardless of state, noise linearly leads to a reduced thalamic response dependency on stimulus frequency. The shaded lines are linear fits.

For this, single TC cells were simulated for extended duration with incoming Poissonian spike trains of 10Hz, modeling a generic input from retinal ganglion cells in-vivo. Here the *retinal* input conductances were fixed. During simulation, for each output spike of the recorded TC cell, the interspike interval (Δ ISI) of the retinal input between the spike which results in the spike response and the preceding one is measured. This way the spike probability or response can be measured as a function of input frequency. Because of the increasingly more rare occurrence of large ISI's (Δ > 400ms) in a Poissonian spike train of 10Hz, the following plots are cut of at 550ms. Until then reasonable long simulation times provide distinguishable uncertainties. Fig 6A shows the results for a TC cell without additional synaptic noise. At resting potential (awake state, $E_L$ = −65mV with $Q_e$ = 14ns) spike response only occurred at summed input spikes with Δ < 50ms with an all-or-none character. At hyperpolarized potential (awake state, $E_L$ = −70mV with $Q_e$ = 24ns) not only input spike summation evoked a response but also ISI's with duration longer than 300ms. These even show higher spike probability compared to spike summation at low ISI's. The difference in input conductances $Q_e$ was necessary to account for equal number of spikes between both states, where the high conductance in the hyperpolarized state captures the effects of T-channels. In the presence of synaptic noise this changes drastically and both TC cells at resting and at hyperpolarized levels exhibit the same spike response, completely independent of input frequency (see Fig 6B). Remarkably, with noise spike probability is significantly lower even with spike summation (ISI→0ms), independent of polarization. These results correctly reproduce the experimental results of [53].

Moving to population level, we present thalamic stimulus response as a function dependent on synaptic noise. Noise acts in a similar way on the frequency dependent response as cortical input. In the same manner, in Fig 6C the slope of the linear response of the TC population as a function of input amplitude is depicted (gain). As with modulating cortical input (refer Fig 4), noise decreases response. However, different to the control of cortical input, where the gain saturates at 0.7Hz$^{-1}$, noise linearly reduces gain until a complete banishment of frequency dependence at very high noise levels ($\tilde{\sigma}_V$ > 12mV). This holds true for all states. In the awake state the loss of gain per membrane fluctuation is (−0.12±0.01)gain/mV. For the sleep state the loss is (−0.028±0.003)gain/mV. Finally, we see that the noise required to equalize the

dependence on frequency between awake and sleep state is significantly higher than for equalizing the dependence on voltage (induced subthreshold fluctuations of 12mV and 4mV, respectively).

## Spindle oscillations

Spindle oscillations are one of the main activity dynamics of the thalamus during NREM sleep or anesthesia [54], strongly influencing the responsiveness of the thalamus in such states. These originate from the superposition of multiple cellular and circuit properties, with especially the mechanism of RE-induced rebound bursts in TC cells in ACh depleted or sleep-like states (see Section F in S1 Appendix for more details).

To enhance this rebound bursting in our sleep state we promote burst firing by adjusting the reset membrane potential ($V_r$) below the sodium spike threshold onset: $V_r = -48$mV for TC and $V_r = -42$mV for RE cell (see [29] for the significant role of $V_r$). This yields sustained burst firing without sustained activation, mimicking T-channel like activation and IPSP barrages in RE cells, which we could not capture with the AdEx sleep state. Accordingly we re-calibrate the mean-field fit to accommodate the change in $V_r$ (Table B in S1 Appendix).

We observe spindles in the proposed models within this adjusted sleep state and when applying an initial kick to evoke activity. Fig 7A and 7B show self sustained oscillations of both full spiking network and mean-field, respectively. Their frequency spectrum and phase space are compared in Fig 7C.

In a previous study [31] only small AdEx networks generated spindles in SR-like dynamics, while at larger scales of $N > 40$ neurons, population activity showed self-driven steady states with AI dynamics. In Fig 7D we show a bifurcation diagram transitioning between connection and balancing synaptic parameters from [31] ($\gamma = 1$) to our parameter values (Table 1, $\gamma = 10$). Increasing connection probability creates a supercritical Andronov-Hopf bifurcation, showing that sufficient connections are necessary for generating and keeping stable spindles at larger network scales. The spindles produced by our mesoscale network show realistic SI dynamics (see Section F in S1 Appendix).

This self-sustained oscillation is remarkably robust in regards to perturbations of all kinds of inputs, producing spindles of same frequency. This renders the thalamus' responsiveness in this spindle-adjusted sleep state highly independent of external input. Only prolonged and constant inputs of a duration longer than multiple spindle periods destroy the synchronization and create steady state AI dynamics. However, spindle oscillations emerge again as soon as the input stops. The bifurcation diagram in connection with [31], shows that thalamic function and responsiveness can be drastically altered depending on specific order parameters, as seen here with connection probability and synaptic conductance.

## Discussion

In this study, we investigated the state-dependent responsiveness of the thalamus at micro to meso-scale. For this we introduced a biologically realistic mean-field model of the thalamus, which captures the population dynamics of thalamocortical relay neurons (TC) and thalamic reticular neurons (RE) in two physiological states: Awake state (high level of ACh neuromodulation, wakefulness and REM sleep) and sleep state (low level of ACh neuromodulation, NREM sleep; see methods) and [9, 42]).

The mean-field model employs the master-equation formalism introduced by El Boustani & Destexhe [47] and incorporates adaptation mechanisms [22]. We constructed it using a bottom-up approach, constraint by existing experiments, following the formalism described by Zerlaut et al. [49], which includes a cell-specific subthreshold-dependent transfer function [50].

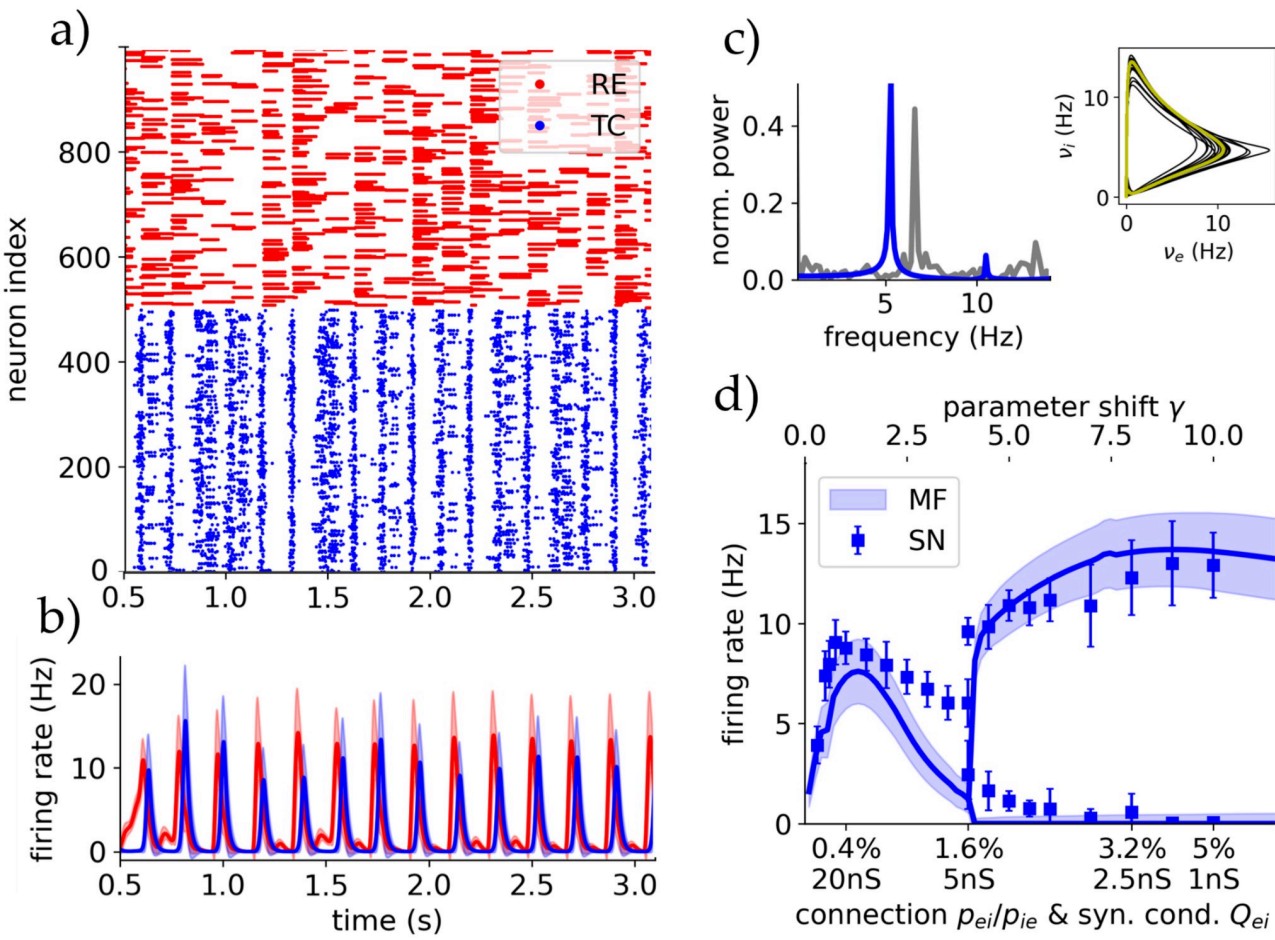

**Fig 7. Spindle oscillations in a sleep-like state generate a highly unresponsive thalamic state. A** Raster plot of the full-scale spiking network with 1000 neurons for spindle parameters (ACh/sleep state with rebound burst). **B** Mean-field oscillations: Firing rates and standard deviations of both TC and RE populations. **C** Fourier spectrum for spiking network (grey) and mean field (blue) of the TC population activity. Inset right: Phase plane in the TC and RE firing rate space. Yellow is the stable limit cycle and black the transient. **D** Bifurcation diagram showing the suggested Andronov-Hopf bifurcation that occurs when gradually increasing the connection probability in the network, for spiking network and mean-field. This corresponds to a parameter shift from the parameters used in [31] with $\gamma = 1$ to the parameters used in this paper with $\gamma = 10$.

Existing mean-field models of thalamocortical dynamics [55, 56], have previously incorporated thalamic single-cell dynamics, such as bursting, into mesoscopic models. While these previous models provide a valuable approach for studying thalamic states, they have included so far only bursting from reticular cells, while bursting from relay neurons was not taken into consideration. In addition, these previous works were focused on reproducing specific types of dynamics (such as slow-waves), while more extensive analysis regarding the transition between thalamic states and the response to sensory stimulation was not investigated. Finally, the key role of neuromodulators in the transition between thalamic neuronal dynamics and states was not captured by these models, which is central to understand the mechanisms behind these transitions. In our work we presented a detailed study reproducing multiple neuronal dynamics found in the thalamus (such as spindles, asynchronous irregular activity and different response modes). Furthermore, these different dynamics emerge as a result of the action of neuromodulators at the single-cell level that are captured within our mean-field model by following a bottom-up approach.

We started by validating our model and showed the mean-field's predictive accuracy through comparison with the spiking network, confirming its ability to replicate the dynamic behavior and population distribution of thalamic cells. We also demonstrated that the mean-field model is capable of predicting the network's subthreshold activity and proved its validity beyond the fitting point. This allows the use of modeling experiments using intracellularly injected currents in combination with this model.

Thalamic responsiveness and it's dependence on internal and external state was then investigated in three steps:

First, we analyzed the important role of bursting in TC cells which provides a mechanism by which the thalamus modulates the transmission of sensory information to the cortex, extending the single cell findings of Sherman & Guillery [7]. We showed that in sleep state response is highly reduced, except for significant (fast changing) stimuli where mainly their timing is transmitted via a strong and fast thalamic response, which is generated by TC cell bursting and delayed inhibition of RE cells. This supports that the thalamus generates and distributes oscillations in NREM sleep states [54]. Additionally, an important validation of the proposed mean-field model is that it captures bursting dynamics, a defining thalamic feature. This is also nontrivial as bursting is highly nonlinear and spike-time dependent, whereas the model is firing rate-based, suggesting interesting future theoretical work.

Then, we examined the influence of external states on thalamic response. We demonstrated that in this model, and in accordance with experiments [6], there is an important distinction in the origin of inputs: sensory-like stimuli experience a more linear response and are therefore transferred more faithfully than cortical-like inputs, which generate a nonlinear response. In sleep-like state the relay of information becomes strongly nonlinear regardless of input origin. Additionally, we identified the modulatory effect of cortical input to (1) repress thalamic response in awake state, via activation of the inhibiting TRN, and (2) to promote a linear response to sensory stimulus in sleep state. (2) would allow the cortex to *wake-up* the thalamus in order to faithfully transfer sensory input, e.g. after a preceding wake-up call of a potentially significant stimulus.

The role of synaptic noise in thalamic response was investigated. The experimental findings of Wolfart [53] were as a first time successfully modeled. We showed that synaptic noise acts as a controller for response also at the population level. The TC cells' step-like response function for single spikes translates well into their collective response at population scale, sharing the same conductance threshold. This allows the thalamus to fine-tune its responsiveness to external stimuli at cell and population level. Additionally, noise diffuses transitions between states of tonic/bursting firing at single cell level and awake/sleep at the population level. We find that in equal manner for single cell and population level, noise banishes the thalamic response dependency on both voltage and frequency. We state the interesting similarity between synaptic noise and cortical input in how both control stimulus transfer and render stimulus response less dependent on stimulus frequency, whose similarity is often only presumed [51]. These insights pronounce the importance of integrating conductance-based subthreshold fluctuations dynamics into meso to macro scale modeling approaches.

Finally, the successful reproduction of spindle-like oscillations in a sleep-like state is an important validation for our thalamic model. We emphasize the necessity of specific substructures within the thalamus for generating realistic oscillations at all scales ([57], Section F in the S1 Appendix). In this state thalamic responsiveness to inputs is highly suppressed. Only strong and prolonged cortical inputs temporarily create AI dynamics during their activation.

In conclusion, our study underscores the value of integrating single-cell dynamics with thalamic specific structure at population-level in understanding the complex role of thalamic responsiveness. With these findings and with offering a biologically realistic and

experimentally grounded mean-field model of the thalamus, which captures the effects of bursting, neuromodulation, and fluctuation, we provide here an essential starting point for: (1) Further investigation of thalamic function and sensory processing. (2) Large-scale modeling (especially the thalamo-cortical loop with already developed cortical mean-fields [22, 34]), while integrating micro-scale cell and synaptic effects with physiological states.

## Supporting information

**S1 Appendix. Supporting information appendix.** Details about employed formulae (Section A), stimuli (Section B), dynamical analysis (Section C), and numerics/software (Section D). Supporting/extending results for the firing adaptation metric (Section E) and for spindle mechanism (Section F). Additional tables and figures.
(PDF)

## Author Contributions

**Conceptualization:** Federico Tesler, Domenico Guarino, Alain Destexhe.

**Funding acquisition:** Alain Destexhe.

**Investigation:** Jorin Overwiening.

**Methodology:** Jorin Overwiening, Federico Tesler, Domenico Guarino.

**Project administration:** Alain Destexhe.

**Software:** Jorin Overwiening.

**Supervision:** Federico Tesler, Domenico Guarino, Alain Destexhe.

**Visualization:** Jorin Overwiening.

**Writing – original draft:** Jorin Overwiening.

**Writing – review & editing:** Jorin Overwiening, Federico Tesler, Domenico Guarino, Alain Destexhe.

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
