## [Decision Letter · Decision Letter 0]

22 Jul 2024

Dear Bsc Overwiening,

Thank you very much for submitting your manuscript "A Multi-Scale Study of Thalamic State-Dependent Responsiveness" for consideration at PLOS Computational Biology. As with all papers reviewed by the journal, your manuscript was reviewed by members of the editorial board and by several independent reviewers. The reviewers appreciated the attention to an important topic. Based on the reviews, we are likely to accept this manuscript for publication, providing that you modify the manuscript according to the review recommendations.

Please take into consideration the points raised by Reviewer 1, in particular related to providing a more in-depth description of the novelty and possibly making Ref. [46] available in a preprint repository, since the paper seems to rely on some results contained in this paper which does not appear to be available.

Sincerely,

Jonathan David Touboul

Academic Editor

PLOS Computational Biology

Daniele Marinazzo

Section Editor

PLOS Computational Biology

Please consider the concerns of Reviewer 1 on providing a more in-depth description of the novelty and possibly making Ref. [46] available in a preprint repository, since the paper seems to rely on some results contained in this paper which does not appear to be available.

Reviewer's Responses to Questions

**Comments to the Authors:**

Reviewer #1: In “A Multi-Scale Study of Thalamic State-Dependent Responsiveness” by Overwiening, Tesler, Guarino & Destexhe, the authors present a biological plausible spiking model of the thalamus and a corresponding mean-field method to capture the behavior of the model. The network consists of inhibitory (RE) and excitatory (TC) cells reciprocally coupled with RE self-inhibiting. This modeling study is able to show a variety of known dynamics in these neural networks, including: sleep & awake states with diff params (Table 1) mimicking presence/absence of ACh, linear/nonlinear response of RE depending on source (stim or crtx) and sleep/awake, and effects of synaptic noise to make sleep/wake response less distinguishable. The work is interesting for capturing a lot of thalamic neural activity. They use a mean-field method that requires fitting the transfer function F to simulations of the underlying model in this case (one could use data), but appears to capture many of the details of the simulated model as shown in figures throughout the paper, but appears to be crucial for describing the supercritical Andronov-Hopf bif in the spindle oscillation section (Fig 7d) — is this true?

Major:

1) I am not fully convinced that this mean-field treatment of these models is ‘novel’ (lines 29, 109). Perhaps the application is novel or the fitting procedure to get the effective transfer function is novel but there are an abundance of mean-field type models for recurrent E&I networks, especially when the units are all intrinsically homogeneous, even when there is (slow) adaptation. An easy fix is just getting rid of the ‘novel’ phrase in these 2 places; if the authors insist on keeping ‘novel’ then a long discussion needs to be added for how different this is compared to [22, 47, 50] and the many other related papers.

2) Related to the prior point. [46] is referenced a lot in the Methods but I can’t seem to find it. Please make it publicly available with a link to arXiv or bioRxiv. There is a worry that this paper is only tweaking 2 parameters compared to [46]: the E_L and the b (spike adapt param). Is that the main methodological difference? I am not discounting the biological results that are captured with different parameters as stated in the paper, but the differences between [46] and this paper need to be clearer in this paper, and if appropriate in [46]. In [46] based on the title, the authors are analyzing different neural network responses with and without ACh, which is the main thrust of this paper.

3) For the mean-field fitting procedure, it all seems a bit strange i.e., the quadratic form of V_theta^eff in eqn (9). Was the exact same procedure used in [50], or were there differences in the model? Presumably this is just a least-squares fit to the transfer function F that is numerically calculated in the network simulations (mapping grid of nu_e,nu_i inputs to outputs).

4) Figure 3a), bottom panel. Where is the red curve, mean-field of RE? For both Fig 3a) b) bottom panel (populations), why isn’t the simulated network of RE shown? The simulated network of TC is shown in blue.

5) Fig 6b is never discussed or referenced in the main text? Is this around lines 550-553? Also, please explain the statement “Remarkably, spike probability is significantly lower than without noise” on line 553, because if this is comparing Fig 6a and 6b, this statement appears true with reasonable ISI (<450ms. is that right?).

6) line 127-129. suggest changing ‘numerously’ to ‘has often’. Can also add the reference “how good are neuron models?” 2009 Science by Gerstner & Naud which was actually fit to LGN data driven by visual stim.

Minor:

line 39 in abstract, what does “drastically changing its responsiveness” refer to? presumably TC cell firing and/or its transfer function, please clarify.

Line 91: not sure what `far of the fitting point’ refers to, I can guess it means that the phenom. models cannot be used to gain insights when params are varied. Please clarify.

Equation (6), cov of 2 pops, but of what? Voltage, W, or firing? Please clarify on line 214. Figured it out after seeing Fig 2e upper right panel & supplementary section S.1 ‘Mean-field correlation’ but it was annoying to hunt it down.

Line 284: ‘of’ -> ‘from’

Figure 2 e), if the non-zero width of the distributions in firing & voltage stem from different (random) number of input connections, please state this. My reading of the paper is that the intrinsic params are homogeneous.

When using multi-scale throughout the paper. What is multi-scale? Seems like it is single-cell spiking model to larger (homogeneous) population, but in my opinion this is not really multi-scale, this is just usual mean-field.

Table S1, what does <!-- mean? Rows 7, 8.</p

Reviewer #2: In this manuscript, Overwiening et al., present a clear addition to the growing body of multiscale models across sleep/wake states, with a focus on thalamocortical and thalamic reticular hyperpolarisation-mediated bursting. They found differences in TC output across varying levels of adaptation, sources of drive, and noise. Along with a demonstration of spindle-like activity, via TC/RE interactions. The study is informative and will be of interest to computational modellers and neuroscientists interested in sleep dynamics. Below I outline some very minor comments.

- I suggest incorporating supp S.5 into the main text.

- Clarify how the bifurcation observed in spindles, identified as a Hopf?

- What is the timestep of cellular simulations?

- “The single cell traces in (a) and (b) reproduce the experiments of Sherman and Guillery [7, ch. 6]. “ could this validation be demonstrated/with reproduced figures in fig 1/2

- Line 597: “create steady state AI dynamics, with how- ever spindle oscillations starting as soon as the input stops.” – unclear

- Line 629: “we state as an important validation that the mean-field model is successfully capturing the important nonlinear thalamic feature of bursting. “ - unclear

- Fig. 3 caption “Fig. 3d, there is depicted the peak response amplitude of the thalamus as a function of” Fig. 3e?

- Reference to fig 6.b is missing in text. A related overall comment, using linking descriptors (e.g. Fig. 6c red/ 6c green) more often would improve the readability of the ms.

- Fig 7 caption, repeated (d) entries.

**Have the authors made all data and (if applicable) computational code underlying the findings in their manuscript fully available?**

Reviewer #1: **No: **I would like the authors to make their code publicly available, in particular the fitting procedure for the transfer function F could be a major detail

Reviewer #2: Yes

PLOS authors have the option to publish the peer review history of their article (what does this mean?). If published, this will include your full peer review and any attached files.

Reviewer #1: No

Reviewer #2: **Yes: **Brandon Munn

Figure Files:

Data Requirements:

Reproducibility:

References:

---

## [Decision Letter · Decision Letter 1]

2 Nov 2024

PCOMPBIOL-D-24-01041R1A multi-scale study of thalamic state-dependent responsivenessPLOS Computational Biology Dear Dr. Overwiening, Thank you for submitting your manuscript to PLOS Computational Biology. Both reviewers are satisfied with your revision, and I expect to recommend publication of your paper. Reviewer #2 had two minor suggestions, including a recommendation of an additional reference to cite. Therefore, instead of accepting immediately your paper, I wanted to give you the opportunity to consider the recommendations of the Reviewer and give you a possibility to revise your manuscript accordingly.  Please submit your revised manuscript within 30 days Jan 02 2025 11:59PM. If you will need more time than this to complete your revisions, please reply to this message or contact the journal office at ploscompbiol@plos.org. Please include the following items when submitting your revised manuscript:*
A rebuttal letter that responds to each point raised by the editor and reviewer(s). You should upload this letter as a separate file labeled 'Response to Reviewers'. This file does not need to include responses to formatting updates and technical items listed in the 'Journal Requirements' section below.*
A marked-up copy of your manuscript that highlights changes made to the original version. You should upload this as a separate file labeled 'Revised Manuscript with Track Changes'.*
An unmarked version of your revised paper without tracked changes. You should upload this as a separate file labeled 'Manuscript'. If you would like to make changes to your financial disclosure, competing interests statement, or data availability statement, please make these updates within the submission form at the time of resubmission. Guidelines for resubmitting your figure files are available below the reviewer comments at the end of this letter. We look forward to receiving your revised manuscript. Kind regards, Jonathan David TouboulAcademic EditorPLOS Computational Biology Daniele MarinazzoSection EditorPLOS Computational Biology

Feilim Mac Gabhann

Editor-in-Chief

PLOS Computational Biology

Jason Papin

Editor-in-Chief

PLOS Computational Biology

 **Journal Requirements:** **Additional Editor Comments (if provided):****Reviewers' comments:** Reviewer's Responses to Questions

**Comments to the Authors:**

Reviewer #1: Very good, thank you.

Reviewer #2: The authors have addressed my technical concerns.

My final concern remains with the use of the phrase novel, it is simply not needed and should be removed from the text/abstract.

The work is a nice exploration without needing to claim novelty, and it only detracts from the work ignoring previous mean-field implementations of thalamocortical dynamics (unless you add a detailed paragraph unpacking the precise novelty). For example, here is a mean field thalamocortical model from a decade ago with thalamic bursting (https://doi.org/10.1016/j.jtbi.2015.01.028) that should be referenced.

**Have the authors made all data and (if applicable) computational code underlying the findings in their manuscript fully available?**

Reviewer #1: None

Reviewer #2: Yes

PLOS authors have the option to publish the peer review history of their article (what does this mean?). If published, this will include your full peer review and any attached files.

Reviewer #1: No

Reviewer #2: **Yes: **Brandon Munn

---

## [Editor Report · Decision Letter 2]

27 Nov 2024

Dear Bsc Overwiening,

We are pleased to inform you that your manuscript 'A multi-scale study of thalamic state-dependent responsiveness' has been provisionally accepted for publication in PLOS Computational Biology.

Best regards,

Jonathan David Touboul

Academic Editor

PLOS Computational Biology

Daniele Marinazzo

Section Editor

PLOS Computational Biology

Feilim Mac Gabhann

Editor-in-Chief

PLOS Computational Biology

Jason Papin

Editor-in-Chief

PLOS Computational Biology

---

## [Editor Report · Acceptance letter]

3 Dec 2024

PCOMPBIOL-D-24-01041R2 

A multi-scale study of thalamic state-dependent responsiveness

Dear Dr Overwiening,

I am pleased to inform you that your manuscript has been formally accepted for publication in PLOS Computational Biology. Your manuscript is now with our production department and you will be notified of the publication date in due course.

With kind regards,

Zsofia Freund
